# Lymph-Venous Anastomosis for Breast Cancer-Related Lymphoedema after Docetaxel-Based Chemotherapy

**DOI:** 10.3390/jcm11051409

**Published:** 2022-03-04

**Authors:** Yuma Fuse, Ryo Karakawa, Tomoyuki Yano, Hidehiko Yoshimatsu

**Affiliations:** Department of Plastic and Reconstructive Surgery, Cancer Institute Hospital of the Japanese Foundation for Cancer Research, Tokyo 135-8550, Japan; yuyuma.fuse@gmail.com (Y.F.); ryo.kyara@gmail.com (R.K.); yanoaprs@icloud.com (T.Y.)

**Keywords:** breast cancer-related lymphedema, lymph-venous anastomosis, docetaxel-based chemotherapy

## Abstract

Docetaxel-based chemotherapy, which is administered before or after axillary lymph node dissection (ALND) in breast cancer patients with positive axillary lymph nodes, is reported as an independent risk factor for development of breast cancer-related lymphoedema (BCRL). Severe hardening of the soft tissue, which is a typical manifestation of BCRL with a history of docetaxel-based chemotherapy, has been considered a contraindication for lymph-venous anastomosis (LVA). This study aimed to evaluate the efficacy of LVA for BCRL with a history of the use of docetaxel. Twenty-six consecutive BCRL patients who underwent LVA were reviewed retrospectively. All patients underwent ALND. Amongst 23 patients who had chemotherapy for breast cancer, docetaxel-based chemotherapy was administered in 12 patients. The postoperative change of the limb circumferences and the improvement of subjective symptoms were assessed. Overall, patients showed improvements of the limb circumferences at the wrist, the elbow, and 5 cm above and below the elbow. There were no statistical differences of the postoperative changes of the circumferences between the docetaxel-administered and non-administered groups (0.25% vs. 2.8% at 5 cm above the elbow (*p* = 0.23), −0.4% vs. 0.7% at 5 cm below the elbow (*p* = 0.56), and 2.5% vs. 2.5 % at the wrist (*p* = 0.82)). LVA is comparably effective for lymphedematous patients who had undergone docetaxel-based chemotherapy before or after ALND.

## 1. Introduction

Breast cancer-related lymphoedema (BCRL), reported to affect 20–45% of breast cancer patients, has a detrimental impact on patients’ quality of life (QOL) [1,2]. The risk factors include axillary lymph node dissection (ALND) and radiotherapy (RT), resulting in the damage of the lymphatic channel [3,4]. Although controversial, docetaxel-based chemotherapy is also regarded as an independent risk factor for BCRL. This regimen is commonly administered before or after ALND in breast cancer patients with positive axillary lymph nodes [5,6]. One adverse effect is fluid accumulation in the arm, and besides it, lymphoedema develops, especially when docetaxel is employed [7]. 

A current gold standard for the treatment of lymphoedema is lymph-venous anastomosis (LVA), in which the lymphatic vessel is anastomosed to a nearby vein [8]. Finding the lymphatic vessels that still retain their function is essential for successful LVA. However, hardening of the soft tissue, which is a typical manifestation in a patient with a history of docetaxel-based chemotherapy, has been considered a contraindication for LVA. This is because the cytotoxicity of docetaxel inflicts severe damage on the lymphatic vessels, resulting in non-functional lymphatic vessels [9]. However, as reported recently, the lymphatic vessels often remain functional in advanced-stage lymphoedema, and the indication of LVA has been widened [10]. In this context, we could hypothesize that LVA is effective for lymphoedematous limbs with a history of docetaxel-based chemotherapy if we can use functional lymphatic vessels. To the best of our knowledge, there have been no reports focusing on the efficacy of LVA in patients with a history of docetaxel-based chemotherapy. 

The aim of the study was to evaluate the efficacy of LVA for BCRL in patients with docetaxel-based chemotherapy. We performed LVA for BCRL patients with or without a history of docetaxel use, and compared the results.

## 2. Materials and Methods

After the approval of the Institutional Review Board at the Cancer Institute Hospital (2021-GB-047), we retrospectively identified 26 consecutive patients with BCRL who underwent LVA by senior surgeons (HY, RK, and YF) from January 2018 through May 2020. Patients with a history of other types of lymphoedema surgery such as lymph node transfer were excluded from the study. The minimum follow-up was six months. The collected information included patients’ demographics, history of breast cancer treatments (surgery, radiotherapy, and chemotherapy), duration of lymphoedema, the International Society of Lymphology (ISL) stage, limb measurements, operative details, and subjective symptoms. Patients’ lymphoedema was classified using the ISL classification: stage 0, stage 1, stage 2a, stage 2b, and stage 3 [11]. A certified lymphoedema therapist evaluated the subjective and quantitative condition before LVA and every 6 months after the operation. All the patients underwent compression therapy by certified therapists before and after the operation. The patients were asked whether their condition of oedema such as heaviness and tightness improved at every check-up.

The circumferences were measured at the wrist, and 5 cm above and below the elbow. The circumferential difference was evaluated as previously reported [12]: the circumference of the unaffected arm was subtracted from that of the affected arm, and subsequently divided by the circumference of the unaffected arm.

### 2.1. Surgical Procedure

All procedures were performed under local anaesthesia except in one case where the procedures were performed under general anaesthesia. Preoperatively, indocyanine green (ICG) was injected at the first and fourth web, medial and lateral to the palmaris longus tendon at the wrist [13]. The lymphatic vessels were marked immediately after the injections under fluorescent observation. The sizable veins were marked using an ultrasonography device. Skin incisions were made under microscopic magnification and the lymphatic vessel and the vein were identified. The bypass was created in an end-to-end fashion. The use of compression garment was resumed immediately after the operation.

### 2.2. Statistical Analysis

Student’s *t*-test or Mann–Whitney U test, and Fisher’s exact test were conducted to analyse the continuous and categoric variables, respectively. *p* values less than 0.05 were considered statistically significant. R v. 4.0.2 (R Foundation for Statistical Computing, Vienna, Austria) was used for data analyses.

## 3. Results

A summary of the patients’ demographics is shown in Table 1. Among the 26 BCRL patients, all patients underwent ALND (100%), and 18 had postoperative radiotherapy (69.2%). Six patients had LVA twice or more during the study period (docetaxel group: 2 patients, non-administered group: 4 patients, *p* = 0.64). Twenty-three patients underwent chemotherapy, and among them, docetaxel was administered in 12 patients. There were no statistical differences in BMI, ISL stage, and the rate of radiotherapy between docetaxel-administered and non-administered patients. The duration of lymphoedema was longer in the docetaxel-non-administered group.

The improvement of the circumference difference was 2.7% (SD: 6.1), 1.6% (SD: 5.2), and 3.7% (SD: 4.1) at 5 cm above and below the elbow and at the wrist, respectively, in the overall population at a postoperative follow-up (mean: 13.8 months) (Table 2 and Figure 1). 

The majority of the patients (*n* = 24, 92%) felt the arm was softer or smaller at the 6-month visit. Between the docetaxel and non-docetaxel groups, there was no statistical difference of the reduction in the circumference difference (0.25% vs. 2.5% at 5 cm above the elbow (*p* = 0.27), 1.3% vs. 1.9% at 5 cm below the elbow (*p* = 0.79), and 3.2% vs. 4.0 % at the wrist (*p* = 0.70)). There was no statistical difference of the improvement of subjective symptoms between both groups (*n* = 19 vs. 9, *p* = 1.00).

## 4. Discussion

The present study demonstrated that LVA was equally effective for patients with a history of docetaxel-based chemotherapy. The circumferential improvement by LVA was not statistically different between the docetaxel-administered and non-administered groups. Subjective improvement was noted in most patients.

In this study, all patients underwent axillary lymph node dissection with mastectomy. Seventy percent of the patients underwent postoperative radiotherapy. Radiotherapy can impair the lymphatic channel in the axilla. Besides, by trapping the draining veins in the radiation-exposed scar, this deteriorates the fluid accumulation of the arm.

Docetaxel-based adjuvant chemotherapy improves overall survival and reduces tumour recurrence in operable breast cancer patients [5]. However, the use of docetaxel leads to some side effects such as fluid retention in extremities. Patients receiving docetaxel have a higher risk of oedema than those having docetaxel-free chemotherapy [14]. This oedema mainly results from abundant extracellular fluid.

There is still controversy surrounding a significant association between docetaxel-based chemotherapy and the development of lymphoedema [7,15,16,17]. Since patients who have had docetaxel-based chemotherapy undergo multiple treatments, it is difficult to assess independent effects of docetaxel on the development of lymphoedema. However, as several studies support the idea that docetaxel can cause lymphoedema, we should not neglect its contribution in the management of lymphoedema [18]. In our experience, the lymphatic vessel was often severely sclerotic in the lymphoedematous limb after the use of docetaxel (Figure 2). This could have been caused by docetaxel leakage in the interstitial tissue, directly damaging the lymphatic vessels. As an in vitro study indicated, docetaxel damages the lymphatic endothelial cells and impairs the lymphatic function [9]. 

While improvements could be seen in the circumference of the upper arm and the wrist, the improvement of the forearm circumference was limited in the docetaxel group. This might be because the fat tissue is likely to accumulate in the medial forearm region after the administration of docetaxel. Magnetic resonance lymphography can be useful to distinguish the accumulation of the fluid and the fat tissue [19]. 

Although no statistical difference could be seen, the overall improvement in the upper arm seemed insufficient in the docetaxel-administered group. A study with a large number of patients could clarify whether the history of docetaxel could affect the efficacy of LVA in certain regions. However, apart from whether LVA worked for the whole arm or the part of the arm, this study still underlines the efficacy of LVA for patients with prior docetaxel administration. 

Some patients underwent multiple LVA if some but insufficient improvement in the lymphoedema status were achieved by the first LVA. We often perform LVA more than two times because all lymphatic drainage routes are likely to be damaged in lymphoedematous patients.

Recently, the efficacy of the Lymphatic Microsurgical Preventive Healing Approach (LYMPHA), in which the lymphatic vessels draining the arm are bypassed immediately at the site of ALND, has been attracting attention [20]. Johnson et al. reported that LYMPHA also prevented BCRL after docetaxel-based chemotherapy [18]. Although reports on the immediate lymphatic reconstruction are still limited, this procedure may be promising for breast cancer patients.

### Limitations

The major limitations are that the number of patients was small. This might have adulterated the interpretation of statistics. A larger sample size could compromise our conclusion. Multiple measurement modalities were not used in postoperative follow-ups.

## 5. Conclusions

The present study demonstrated that LVA is effective even for lymphedematous patients who underwent docetaxel-based chemotherapy before or after ALND. The circumference difference reduced postoperatively and subjective symptoms improved in most patients.

## Figures and Tables

**Figure 1 jcm-11-01409-f001:**
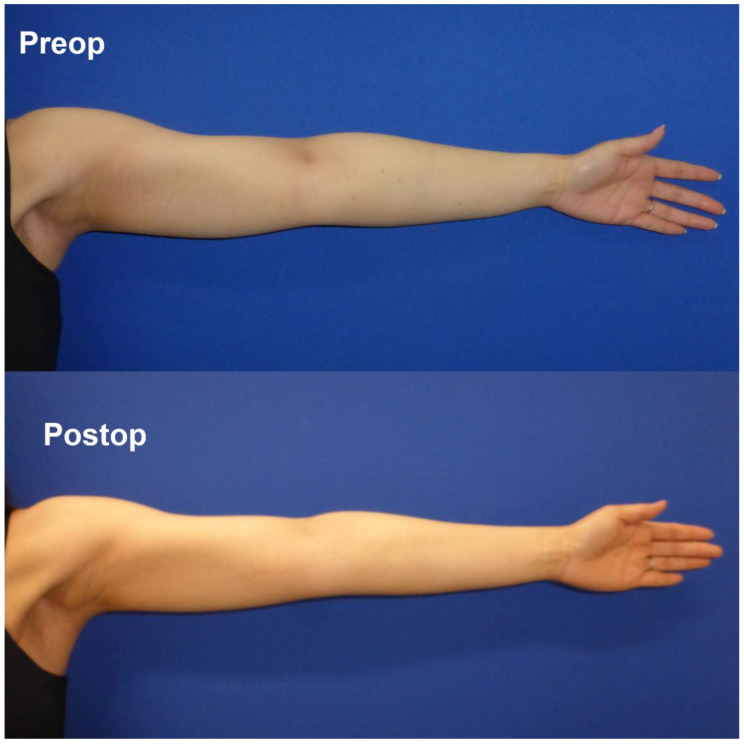
Pre- and postoperative views of a docetaxel-administered patient. Oedema in the forearm and the wrist improved with the contour of the tendon more noticeable.

**Figure 2 jcm-11-01409-f002:**
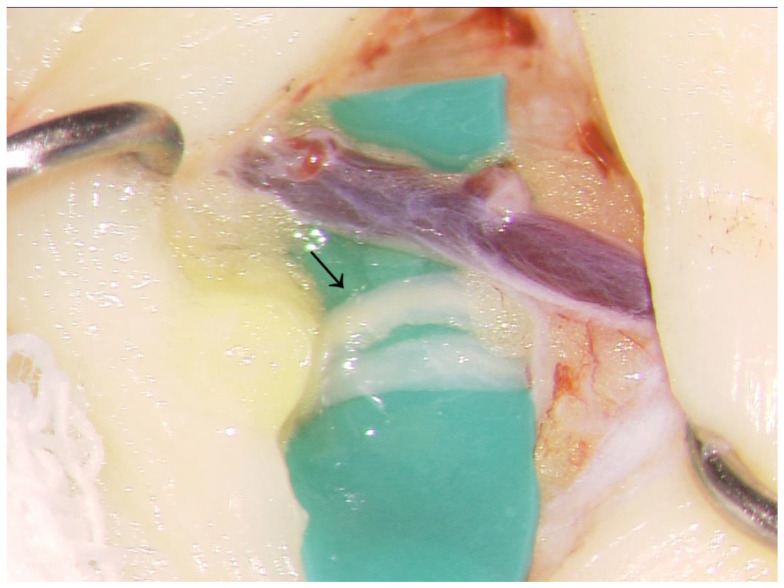
Intraoperative view of degenerated lymphatic vessels. The lymphatic vessels (arrow head) were severely sclerotic.

**Table 1 jcm-11-01409-t001:** Patient demographics.

	Docetaxel Administered	Non-Administered	*p* Value
*n*	12	11	
Age (mean (SD))	52.83 (7.93)	60.18 (12.44)	0.103
BMI (mean (SD))	22.84 (4.38)	22.18 (2.09)	0.654
ISL stage ^1^ (%)			0.152
0	0 (0.0)	0 (0.0)	
1	0 (0.0)	2 (18.2)	
2a	12 (100.0)	8 (72.7)	
2b	0 (0.0)	1 (9.1)	
3	0 (0.0)	0 (0.0)	
Lymphadenectomy	12 (100.0)	11 (100.0)	NA
Radiotherapy	9 (75.0)	7 (63.6)	0.89
Duration, month (mean (SD))	37.50 (24.47)	88.27 (52.11)	0.006
Follow-up period, month (median (IQR))	8.5 (6.0, 14.5)	12.0 (10.0, 19.8)	0.14

^1^ ISL: International Society of Lymphology.

**Table 2 jcm-11-01409-t002:** The postoperative change of the bilateral difference of the limb circumference and the improvement of subjective symptoms.

	Docetaxel Administered	Non-Administered	*p* Value
Change of Circumference Difference, % (median, (IQR))			
>5 cm elbow	−0.25 [−3.35, 2.25]	−2.78 [−8.14, −1.87]	0.225
<5 cm elbow	0.37 [−3.67, 2.84]	−0.74 [−4.07, 2.31]	0.564
wrist	−2.45 [−6.22, 0.27]	−2.54 [−6.40, −0.75]	0.817
subjective symptoms improved, *n* (%)	10 (90.9)	9 (81.8)	1

## Data Availability

The data presented in this study are available on request from the corresponding author, H.Y.

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
