# Peer review of "Lymph-Venous Anastomosis for Breast Cancer-Related Lymphoedema after Docetaxel-Based Chemotherapy"

_jcm, 2022, doi:10.3390/jcm11051409_

Round 1

Reviewer 1 Report

well conducted, interesting work that deals with a very useful and underestimated topic. unfortunately, the sample size il quite low and this impacts on the significance of content.

the abstract needs to be improved, in particular line 15: replace underwent with had undergone. line 18: after LVA, ovewrall patients .... indeed at the current status the abstract doen't appear sufficient clear

line 97: specify the period of the follow-up 

fig 2: the arrow head doesn't result adequate. use the simple arrow

Author Response

1. well conducted, interesting work that deals with a very useful and underestimated topic. unfortunately, the sample size il quite low and this impacts on the significance of content.

Response: Thank you for your feedback. The small sample size is a significant limitation and we described this in Discussion.

2. the abstract needs to be improved, in particular line 15: replace underwent with had undergone. line 18: after LVA, ovewrall patients .... indeed at the current status the abstract doen't appear sufficient clear

Response: Thank you for the comment. We revised the abstract.

3. line 97: specify the period of the follow-up 

Response: Thank you for the comment. We added a median of the follow-up period with the IQR in the result section (docetaxel-group: 8.5 months (IQR: 6.0-14.5), control: 12 months (IQR: 10.0-19.8). There was no statistical difference between the groups (p=0.14).

4. fig 2: the arrow head doesn't result adequate. use the simple arrow

Response: Thank you for the comment. We replaced the arrow head with a simple arrow.

Reviewer 2 Report

Reduction of limb volume in patients who have developed lymphoedema secondary to treatment for breast cancer is important.  Developing the evidence base around the use of surgical intervention for lymphoedema is particularly critical.  This manuscript will add to this base; however, a few changes are required to make it more clear.

  • I am uncomfortable with the lymphoedema the patients are experiencing being labelled “docetaxel-related lymphoedema”.  These patients underwent ALND as well as, for many but not all, radiotherapy, which are both also risk factors for the development.  Its inaccurate to say this is only related to docetaxel.  Much of the controversy on whether or not taxane based chemotherapys are a risk factor for lymphoedema or not comes from studies that include both patients who have undergone ALND and SNB.  In studies where these are separated, ALND comes out as a risk factor only for the ALND group (e.g. Kilbreath et al, 2016, Breast).  Either way, these patients all have undergone multiple treatments for their breast cancer that have put them at risk for lymphoedema, including but not limited to the docetaxel and it is therefore inaccurate to classify their lymphoedema as docetaxel-related.
  • In the introduction, more support for the aim is needed. I’m unclear as to why there would be a thought that patients who have had docetaxel at some point in their treatment would be less responsive to an LVA procedure.  The reasoning for this assumption needs to be better explored.

  • More information is needed to better help the reader understand the timing of the surgery relative to both the development of lymphoedema and the patient undergoing the docetaxel related lymphoedema. Why did some patients undergo multiple LVA surgeries and what impact did this have on the change in circumference?  Did this group have a longer or shorter follow up or mixed?  For those with longer follow up or multiple follow ups, which measurement did you use for the analysis and was this standardised?  When did the LVA procedures happen relative to the docetaxel; when was the lymphoedema diagnosed relative to the docetaxel and the surgeries?

  • The results related to the circumference changes would be better presented as a range because the standard deviation is quite large. While for the whole group there was no difference between the group in terms of improvements between the docetaxel and non-docetaxel group, they are small groups and appear to have large variability and only presenting the mean and standard deviation may be missing important differences between groups if there are any.  Its also not reported if the data is normatively distributed. If its not, median and IQR would be a more appropriate than mean and SD

  • Please provide more details on what subjective symptoms were assessed and how.

  • I don’t understand the limitation around BIS usage as an assessment in this study. Why would fluid that is accumulating (lymphoedema) be difficult to assess?  You’ve referenced the Lee et al study that looked at the time course of edema/ECF volume increases through and after Taxane based chemo elsewhere in your paper.

Author Response

Reviewer2:

  1. Reduction of limb volume in patients who have developed lymphoedema secondary to treatment for breast cancer is important.  Developing the evidence base around the use of surgical intervention for lymphoedema is particularly critical.  This manuscript will add to this base; however, a few changes are required to make it more clear.

Response: Thank you for your feedback.

  1. I am uncomfortable with the lymphoedema the patients are experiencing being labelled “docetaxel-related lymphoedema”.  These patients underwent ALND as well as, for many but not all, radiotherapy, which are both also risk factors for the development.  Its inaccurate to say this is only related to docetaxel.  Much of the controversy on whether or not taxane based chemotherapys are a risk factor for lymphoedema or not comes from studies that include both patients who have undergone ALND and SNB.  In studies where these are separated, ALND comes out as a risk factor only for the ALND group (e.g. Kilbreath et al, 2016, Breast).  Either way, these patients all have undergone multiple treatments for their breast cancer that have put them at risk for lymphoedema, including but not limited to the docetaxel and it is therefore inaccurate to classify their lymphoedema as docetaxel-related.

Response: Thank you for the comment. As the reviewer mentioned, there are still controversy whether docetaxel is a causative factor of lymphoedema. Since docetaxel is conventionally administered for lymph node-positive breast cancer, it is difficult to assess independent effects of docetaxel on lymphoedema. However, some surgeons have focused on the effects of docetaxel when evaluating the efficacy of LVA. The aim of the present study was not to discuss docetaxel caused lymphoedema but to evaluate the efficacy of LVA for patients with a history of docetaxel administration. This point was clarified in Discussion.

We do fully agree that the term “docetaxel-related lymphoedema” can cause confusion, and thus replaced the expression with “BCRL with a history of docetaxel-based chemotherapy” throughout the manuscript.

  1. In the introduction, more support for the aim is needed. I’m unclear as to why there would be a thought that patients who have had docetaxel at some point in their treatment would be less responsive to an LVA procedure.  The reasoning for this assumption needs to be better explored.

 Response: Thank you for the comment. Docetaxel impairs the lymphatic endothelial cells and deteriorate the lymphatic function (Am W, Je B, H P, et al. Docetaxel causes lymphatic endothelial cell apoptosis and impairs lymphatic function and gene expression in vitro. J Transl Sci. 2021;7(2)). Since the efficacy of LVA depends on the remaining function of the lymphatic vessels, some surgeons hesitate to perform LVA for patients with docetaxel-related lymphoedema. We revised the introduction as follows:

‘A current gold standard for the treatment of lymphoedema is lymph-venous anastomosis (LVA) in which the lymphatic vessel is anastomosed to a nearby vein. Finding the lymphatic vessels that still retain their function is essential for successful LVA. However, severe hardening of the soft tissue, which is a typical manifestation of in a patient with a history of docetaxel-based chemotherapy, has been considered a contraindication for LVA. This is because cytotoxicity of docetaxel inflicts severe damage to the lymphatic vessels, resulting in non-functional lymphatic vessels.’

  1. More information is needed to better help the reader understand the timing of the surgery relative to both the development of lymphoedema and the patient undergoing the docetaxel related lymphoedema. Why did some patients undergo multiple LVA surgeries and what impact did this have on the change in circumference?  Did this group have a longer or shorter follow up or mixed?  For those with longer follow up or multiple follow ups, which measurement did you use for the analysis and was this standardised?  When did the LVA procedures happen relative to the docetaxel; when was the lymphoedema diagnosed relative to the docetaxel and the surgeries?

Response: Thank you for the comment. We added a median of the follow-up period with the IQR in the result section (docetaxel-group: 8.5 months (IQR: 6.0-14.5), control: 12 months (IQR: 10.0-19.8). There was no statistical difference between the groups (p=0.14).

Diagnosis of the disease was made by certified lymphoedema therapists during the follow-up after breast cancer surgery. The patients were referred to our lymphoedema clinic when they complained arm oedema. All the diagnosis was made after docetaxel administration. We added this to the result section.

Some patients had multiple LVAs. Second LVA was offered to patients with some but insufficient improvement in their lymphoedema status. We often perform LVA more than two times or more because all lymphatic drainage routes are likely to be damaged in lymphoedematous patients. In our practice, if the first LVA was not successful, we move on to the vascularized lymph node transfer. In the present study, LVA was efficacious in all patients. No difference of the number of patients who had multiple LVAs were observed between the groups. We revised the Results and Discussion as follows:

‘Six patients had LVA twice or more during the study period (docetaxel group: 2 patients, non-administered group: 4 patients, p=0.64)’.

‘Some patients underwent multiple LVA if some but insufficient improvement in the lymphoedema status were achieved by the first LVA. We often perform LVA more than two times because all lymphatic drainage routes are likely to be damaged in lymphoedematous patients.’

5.The results related to the circumference changes would be better presented as a range because the standard deviation is quite large. While for the whole group there was no difference between the group in terms of improvements between the docetaxel and non-docetaxel group, they are small groups and appear to have large variability and only presenting the mean and standard deviation may be missing important differences between groups if there are any.  Its also not reported if the data is normatively distributed. If its not, median and IQR would be a more appropriate than mean and SD

Response:  Thank you for the comment. We replaced the mean and SD with a median and IQR for each result of the circumference change. There were also no differences between the groups on Mann-Whitney test.

  1. Please provide more details on what subjective symptoms were assessed and how.

 Response: We surveyed the improvement of subjective symptoms (feeling smaller, softer, and painless) at every check-up. We revised the Method and Results as follows:

‘The patients were asked about their condition of oedema such as heaviness and tightness at every check-up.’

7: I don’t understand the limitation around BIS usage as an assessment in this study. Why would fluid that is accumulating (lymphoedema) be difficult to assess?  You’ve referenced the Lee et al study that looked at the time course of edema/ECF volume increases through and after Taxane based chemo elsewhere in your paper.

Response: Thank you for the comment. We removed the description on BIS.

Reviewer 3 Report

Thank you for submitting an interesting article to tis journal.

In gerneral, the manuscript is well written. However, I do not think that this mansucript introduces impactful new information or techniques to the readership.

Author Response

Thank you for submitting an interesting article to tis journal.

In general, the manuscript is well written. However, I do not think that this manuscript introduces impactful new information or techniques to the readership.

Response: Thank you for the comment. The effects of docetaxel on lymphedema treatment had not been discussed in detail. We believe the present study will help the development of lymphedema treatment.

Round 2

Reviewer 1 Report

no further indications

Author Response

Thank you for further review our manuscript.

Reviewer 2 Report

Thank you for your revisions.  They have improved the quality and clarity of the manuscript.

A few minor suggestions for fixing:

Pg 1, line 21-22 (0.25% vs 2.8% at 5 cm above the elbow (p=0.23), -0.4% vs 0.7% at 5 cm below the elbow (p=0.56), and 2.5% vs 2.5 % at the wrist (p=0.82).

  • Missing additional ) at the end of the sentence

Pg 2 lines 65-66: “The patients were asked about the condition of oedema such as heaviness and tightness at every check-up.”

  • Was this a VAS scale or just asking them if it was better or worse? This information is needed to give context to the details in table 2 about symptom improvement.

Table 1- need 1 superscript next to ISL in table 1 column

Author Response

  1. Pg 1, line 21-22 (0.25% vs 2.8% at 5 cm above the elbow (p=0.23), -0.4% vs 0.7% at 5 cm below the elbow (p=0.56), and 2.5% vs 2.5 % at the wrist (p=0.82).
  • Missing additional ) at the end of the sentence

Response: Thank you for the comment. We added a bracket.

  1. Pg 2 lines 65-66: “The patients were asked about the condition of oedema such as heaviness and tightness at every check-up.”
  • Was this a VAS scale or just asking them if it was better or worse? This information is needed to give context to the details in table 2 about symptom improvement.

Response: Thank you for the comment. The patients were asked whether their condition of oedema improved or not at every check-up.

  1. Table 1- need 1 superscript next to ISL in table 1 column

Response: Thank you for the comment. We added a superscript.